



# Quantification of the modelling uncertainties in atmospheric release source assessment and application to the reconstruction of the autumn 2017 Ruthenium 106 source

Joffrey Dumont Le Brazidec[1,2], Marc Bocquet[2], Olivier Saunier[1], and Yelva Roustan[2]

[1]IRSN, PSE-SANTE, SESUC, BMCA, Fontenay-aux-Roses, France
[2]CEREA, Joint laboratory École des Ponts ParisTech and EDF R&D, Université Paris-Est, Marne-la-Vallée, France

**Correspondence:** Joffrey Dumont Le Brazidec (joffreydumont@hotmail.fr)

**Abstract.**

Using a Bayesian framework in the inverse problem of estimating the source of an atmospheric release of a pollutant has proven fruitful in recent years. Through Markov chain Monte Carlo (MCMC) algorithms, the statistical distribution of the release parameters such as the location, the duration, and the magnitude as well as the likelihood covariances can be sampled

so as to get a complete characterisation of the source. In this study, several approaches are described and applied to improve on these distributions, and therefore to get a better representation of the uncertainties. First, a method based on ensemble forecasting is proposed : physical parameters of both the meteorological fields and the transport model are perturbed to create an enhanced ensemble. In order to account for model errors, the importance of ensemble members are represented by weights and sampled together with the other variables of the source. Secondly, the choice of the statistical likelihood is shown to alter

the nuclear source assessment, and several suited distributions for the errors are advised. Finally, two advanced designs of the covariance matrix associated to the observation error are proposed. These methods are applied to the case of the detection of Ruthenium 106 of unknown origin in Europe in autumn 2017. A posteriori distributions meant to identify the origin of the release, to assess the source term, to quantify the uncertainties associated to the observations and the model, as well as densities of the weights of the perturbed ensemble, are presented.

## 1 Introduction

### 1.1 Bayesian inverse modelling for source assessment

Inverse modelling of nuclear release source is an issue fraught with many uncertainties (Abida and Bocquet, 2009). Therefore variational techniques (Saunier et al., 2013; Bocquet, 2012), which only provide a deterministic and thus unique solution to the problem, miss valuable information such as other potential sources. On the other hand, probabilistic methods develop

stochastic solutions in order to capture all information from the data. In particular, Bayesian methods have proven to be very efficient in source term estimation (STE) problems. Several techniques such as the iterative variational Bayes method (Tichý et al., 2016) tested using data from the European Tracer Experiment (ETEX), or an adaptive scheme based on importance





sampling (Rajaona et al., 2015) and tested on the Fusion Field Trials 2007 experiment, have been developed. Sampling using Markov chain Monte Carlo methods is a very popular technique, since it allows to directly reconstruct the posterior distribution

of the source. It has been applied by Delle Monache et al. (2008) to estimate the Algeciras incident source location. Keats et al. (2007) sampled the source parameters of a complex urban environment with the help of the Metropolis-Hastings algorithm. The emissions of Xenon-133 from the Chalk River Laboratories medical isotope production facility were reconstructed with their location by Yee et al. (2014). Various methods including Markov chain Monte Carlo (MCMC) techniques are used by Liu et al. (2017) to assess the source term of the Chernobyl and Fukushima Daiichi accidents and their associated uncertainties.

## 1.2 Ensemble methods

A major source of uncertainties in inverse modelling for source term estimation of nuclear accidents finds its origin in the meteorological fields and the transport models (Sato et al., 2018) which are used to simulate the plume of the emission. Weather forecast uncertainties arise from errors in the initial conditions and approximation in the construction of the numerical model. They can be evaluated through the use of an ensemble forecast (Leith, 1974). Several realisations of a same forecast are

considered, where for each realisation, the initial condition and the numerical model are perturbed.

An efficient way to use these forecasts to better estimate uncertainties is to combine them. This approach is known as multi-model ensemble forecasting (Zhou and Du, 2010). More specifically, ensemble members can be combined through sequential aggregation, where each member of the ensemble is given a weight. An aggregated forecast is then formed by the weighted linear combination of the forecasts of the ensemble, where the weights are computed using assorted methods such as machine

learning algorithm (Mallet et al., 2009) or least squares algorithms (Mallet and Sportisse, 2006). For weather forecasting, these weights can depend on past observations or analyses (Mallet, 2010). The uncertainty due to the transport model can also be solved through ensemble methods such as Delle Monache and Stull (2003) who use a mix of diverse air quality models.

### 1.3 Probabilistic description of the problem

We wish to parametrise the distribution of the variable vector $\boldsymbol{x}$ describing the source of a release. In the case of a point-wise

source of unknown location, the most important variables describing the source are the coordinates longitude-latitude $(x_1, x_2)$, the vector $\ln \boldsymbol{q}$, where each component corresponds to the logarithm of the release $\boldsymbol{q}$ on a given time interval (e.g., an hour or a day), and the covariance matrix $\mathbf{R}$ describing the uncertainties and defined below. The posterior probability distribution is written with the help of Bayes' theorem as

$$p(\boldsymbol{x}|\boldsymbol{y}) = \frac{p(\boldsymbol{y}|\boldsymbol{x})p(\boldsymbol{x})}{p(\boldsymbol{y})} \propto p(\boldsymbol{y}|\boldsymbol{x})p(\boldsymbol{x}) \tag{1}$$

with $\boldsymbol{y}$ the observation vector, usually a set of air concentration measurements, and $\boldsymbol{x}$ the source variable vector. The first term $p(\boldsymbol{y}|\boldsymbol{x})$ of equation (1) corresponds to the likelihood, the distribution quantifying the fit of a statistical model (here the characterisation of the source $\boldsymbol{x}$) with the data (the observation vector $\boldsymbol{y}$). The second term $p(\boldsymbol{x})$ describes the prior on the source vector. Once the posterior probability distribution is made explicit up to a normalisation constant, several sampling techniques can be applied to it (Liu et al., 2017).





The shape of the posterior distribution strongly depends on the uncertainties related to the modelling choices and the data. The objective of this study is to investigate the various sources of uncertainties compounding the problem of source reconstruction, and to propose solutions to better evaluate them, i.e., to increase our confidence in the posterior distribution reconstructed. The quantification of the uncertainties largely depends on the definition of the likelihood and its components. The choice of the likelihood is the concern of section 2.1. As mentioned above, the likelihood term quantifies the fitness between the measure-
ments $\boldsymbol{y}$ and a given parametrisation of the source $\boldsymbol{x}$. To make these two quantities comparable, a set of modelled concentrations corresponding to the observations are computed. These concentrations, also called predictions, are the results of the dispersion of the source $\boldsymbol{x}$, and are therefore depending on the parametrisation of the transport model and on the meteorological fields.

In this paper and in the case of a source of unknown location, the predictions are written as $\boldsymbol{y}_S = \mathbf{H}_{x_1,x_2}\boldsymbol{q}$ where $\mathbf{H}$ is the observation operator, the matrix representing the resolvent of the atmospheric transport model, and $\mathbf{H}_{x_1,x_2}$ its definition for
a source of coordinates $x_1, x_2$. Therefore, the observation operator does vary linearly in accordance with the release vector. However, $\mathbf{H}$ is not linear in the coordinates. When the coordinates are unknown, they may be investigated in a continuous space. The computation of a matrix $\mathbf{H}_{x_1,x_2}$ for a specific couple of coordinates being expensive, a set of observation operators linked to specific locations is computed on a regular mesh $G$ prior to the Bayesian sampling presented in section 3.2.3. The observation operators are therefore interpolated from the set of observation operators pre-computed on $G$.
Equation (1) can be expanded as

$$p(\boldsymbol{x}|\boldsymbol{y}) \propto p_{\text{likelihood}}(\boldsymbol{y}|\mathbf{H}_{x_1,x_2}(m)\boldsymbol{q}, \mathbf{R})\, p_{\text{prior}}(x_1, x_2, \ln\boldsymbol{q}, \mathbf{R}, ...) \tag{2}$$

where uncertainties are embodied in

- the observations $\boldsymbol{y}$;

- the physical models: the meteorological fields $m$ and the dispersion $\mathbf{H}$;

- the likelihood definition: its choice and the design of its associated error covariance matrix $\mathbf{R}$;

- the representation error: the release rates $\boldsymbol{q}$ as a discrete vector (while the release is a continuous phenomenon), the observation operator $\mathbf{H}_{x_1,x_2}$ as an interpolation, and the observations for which the corresponding predictions are calculated in a mesh containing them;

- the choice of the priors.

In this paper, we focus on the uncertainties emanating from the physical models and the definition of the likelihood.

## 1.4   Objectives of this study

This study is a continuation from a previous study from the authors (Dumont Le Brazidec et al., 2020). It aims at exploring the various sources of uncertainty which are compounding the inverse problem and proposing solutions to better evaluate them: three key issues are investigated. First, in section 2.1, we investigate the design of the likelihood distribution, which is the main





ingredient in the definition of the posterior distribution. Secondly, we propose two new designs of the likelihood covariance matrix to better evaluate errors in section 2.2. Finally, in section 2.3, we describe an ensemble based method for taking into account the uncertainties related to the meteorological fields and atmospheric transport model: we construct $\mathbf{H}$ as a weighted sum of observation operators created out of diverse physical parameters.

Subsequently, the interest of these three propositions is illustrated with an application on the $^{106}$Ru release in September
2017. First, a description of the context, the observation dataset, and the state of the art of the release event is introduced in section 3.1. Then the parametrisation of the physical model is presented in section 3.2.1. Finally, the results of the successive applications of the assorted methods described in section 2, combined or not, are presented in section 3.3. A summary of the various configurations of each application is proposed in section 3.3.1. Conclusions on the contribution of each method are finally proposed.

## 95  2  Evaluating uncertainties in the Bayesian inverse problem

### 2.1  Choice of the likelihood

In the field of source assessment and more precisely, radioactive materials source assessment, most of the likelihoods are defined as Gaussian (Yee, 2008; Winiarek et al., 2012; Saunier et al., 2013; Bardsley et al., 2014; Yee et al., 2014; Winiarek et al., 2014; Rajaona et al., 2015; Tichý et al., 2016) or more recently log-normal (Delle Monache et al., 2008; Liu et al., 2017;
Saunier et al., 2019; Dumont Le Brazidec et al., 2020) or similar to a log-normal (Senocak et al., 2008). The multivariate Gaussian probability density function (pdf), of mean $\mathbf{H}\boldsymbol{x}$ and covariance matrix $\mathbf{R}$, is written as

$$p(\boldsymbol{y}|\boldsymbol{x}) = \frac{1}{\sqrt{2\pi|\mathbf{R}|}}\exp\left(-\frac{(\boldsymbol{y}-\mathbf{H}\boldsymbol{x})^{\top}\mathbf{R}^{-1}(\boldsymbol{y}-\mathbf{H}\boldsymbol{x})}{2}\right). \tag{3}$$

In this section, we assume that the covariance matrix $\mathbf{R}$ is equal to $r\mathbf{I}$ where $r$ is a positive diagonal coefficient. The cost function, i.e., the negative of the log-likelihood, of the Gaussian probability density function (pdf) is written (up to a normalisation
constant) as

$$\mathcal{J}_{\boldsymbol{y}}(\boldsymbol{x}) = -\ln p(\boldsymbol{y}|\boldsymbol{x}) = \frac{\mathrm{N}_{\mathrm{obs}}\ln r}{2} + \frac{(\boldsymbol{y}-\mathbf{H}\boldsymbol{x})^{\top}(\boldsymbol{y}-\mathbf{H}\boldsymbol{x})}{2r} \tag{4}$$

with $\mathrm{N}_{\mathrm{obs}}$ the number of observations of the problem. The cost function is a matter of judgement; it measures how detrimental is a difference between an observation and a prediction. When the observations and the predictions are equal, the likelihood part of the cost should be zero and it should increase when the difference between the observation and the prediction values
grows.

With the assumption $\mathbf{R}=r\mathbf{I}$, choosing Gaussian is equivalent to tremendously favouring high values: the Gaussian cost function value of an observation-prediction couple ($y = 100$ mBq.m$^{-3}$, $y_S = 120$ mBq.m$^{-3}$) is a hundred of times greater than of ($y = 10$ mBq.m$^{-3}$, $y_S = 12$ mBq.m$^{-3}$). In other words, reducing the difference between the observation and the prediction of the first couple is an objective a hundred of times more important for the algorithm than for the second couple.





In other words, with a gaussian likelihood, inverse modelling is dominated by the most significant measurements in real-case studies. Every measurement which will be negligible in magnitude compared to the greatest ones will be negligible in the source retrieval.

We think that the whole measurement set should bring information. More generally, the following inventory lists the criteria that a good likelihood choice for nuclear source assessment should fulfil:

– positive support: should be defined on the semi-infinite interval $[0, +\infty[$ since the observations and predictions are all positive by nature;

– symmetry between the prediction vector and the observation vector, i.e., $p(\boldsymbol{y}; \mathbf{H}\boldsymbol{x}, \mathbf{R}) = p(\mathbf{H}\boldsymbol{x}; \boldsymbol{y}, \mathbf{R})$. The couple $\left(y = 20\mathrm{mBq.m}^{-3}, y_S = 40\mathrm{mBq.m}^{-3}\right)$ should have the same penalty as $\left(y = 40\mathrm{mBq.m}^{-3}, y_S = 20\mathrm{mBq.m}^{-3}\right)$;

– relativity: the ratio of the cost function value of a couple $(20\mathrm{mBq.m}^{-3}, 40\mathrm{mBq.m}^{-3})$ with a couple $(200\mathrm{mBq.m}^{-3}, 400\mathrm{mBq.m}^{-3})$ should be mitigated and close to 1 as a general rule;

– existence of a covariance matrix, and of a term able to play the role of the modelled predictions. Indeed, the likelihood measures the difference between the observations and the predictions, which should therefore appear as a parameter of the distribution. Distributions with a location parameter comply with this requirement.

In particular, three distributions were found to satisfy these diverse criteria: the log-normal distribution already used in several studies, the log-Laplace and the log-Cauchy distributions, which have for cost functions up to a normalisation constant (Satchell and Knight, 2000; McDonald, 2008)

$$\mathcal{J}_{\mathrm{log-normal}}(\boldsymbol{y}; \mathbf{H}\boldsymbol{x}, \mathbf{R}, \boldsymbol{y_t}) = \frac{1}{2}\|\ln(\boldsymbol{y} + \boldsymbol{y_t}) - \ln(\mathbf{H}\boldsymbol{x} + \boldsymbol{y_t})\|_{2,\mathbf{R}^{-1}}^2 + \frac{\mathrm{N_{obs}}}{2}\ln(r), \tag{5a}$$

$$\mathcal{J}_{\mathrm{log-Laplace}}(\boldsymbol{y}; \mathbf{H}\boldsymbol{x}, \mathbf{R}, \boldsymbol{y_t}) = \|\ln(\boldsymbol{y} + \boldsymbol{y_t}) - \ln(\mathbf{H}\boldsymbol{x} + \boldsymbol{y_t})\|_{1,\mathbf{R}^{-1}} + \mathrm{N_{obs}}\ln(r), \tag{5b}$$

$$\mathcal{J}_{\mathrm{log-Cauchy}}(\boldsymbol{y}; \mathbf{H}\boldsymbol{x}, \mathbf{R}, \boldsymbol{y_t}) = \sum_{i=1}^{\mathrm{N_{obs}}} \ln\left(r + (\ln(y_i + y_t) - \ln((\mathbf{H}\boldsymbol{x})_i + y_t))^2\right) - \frac{1}{2}\ln(r), \tag{5c}$$

where $\boldsymbol{y_t}$ is a positive threshold vector to ensure that the logarithm function is defined for zero observations or predictions. The $l_2$ and $l_1$ norms are defined as $\|\boldsymbol{v}\|_2 = \sqrt{\boldsymbol{v}^\top \boldsymbol{v}}$ and $\|\boldsymbol{v}\|_1 = \sum_i |v_i|$, respectively. These three distributions are subsumed by the Generalised Beta Prime (or GB2) (Satchell and Knight, 2000; McDonald, 2008) and share a common point; all three are shaped around the subtraction of the logarithm of the observation by the logarithm of the prediction. Due to the logarithmic property

$\ln\frac{a}{b} = \ln(a) - \ln(b)$, several criteria previously defined are met. First the cost is a function of the ratio of the observation to the prediction. Secondly, a location term appears as $\ln((\mathbf{H}\boldsymbol{x})_i + y_t)$. And finally, with the help of a square or an absolute value, a symmetry between the observation and the prediction is guaranteed. Each of these choices requires a threshold (and even two for the log-Cauchy case, discussed at the end of this section) whose value significantly impacts the results and will be discussed later on.



Their difference lies in the treatment of the relative quantity $\ln(y_i + y_t) - \ln((\mathbf{H}\boldsymbol{x})_i + y_t)$. With the $l_2$ norm, the log-normal drives most of the penalty on the large (relative) differences, and removes almost all penalty from the small differences. With the $l_1$ norm, the log-Laplace curve of the relative quantity is flatter in comparison. This translates in the fact that the inverse modelling will not be sensitive to one couple in particular, even if for this couple the difference between the observation and the prediction is large. The motive therefore to use log-Laplace is to avoid having outliers driving the entire sampling, i.e.,

driving the entire search of the source. The log-Cauchy distribution is the one with the most interesting behaviour and mixes log-normal and log-Laplace natures. The logarithm mitigates the penalty of large differences, but also removes any penalty from the small differences. The rationale of using the log-Cauchy distribution is consequently to avoid outliers, but at the same time to avoid taking into account negligible differences.

For all choices, the value of $y_t$ is crucial to evaluate the penalty on a couple involving a zero observation or prediction.

In other words, it figures how the cost of a zero observation and a non-zero prediction (or the contrary) should compare to a positive couple. We consider that the penalty on a couple $(20 \text{ mBq.m}^{-3}, 0 \text{ mBq.m}^{-3})$ should be a big multiple of the penalty on $(400 \text{ mBq.m}^{-3}, 100 \text{ mBq.m}^{-3})$. As a consequence, it can be deduced that a "good" threshold for the log-normal distribution in a case involving important quantities released should lie between $0.5 \text{ mBq.m}^{-3}$ and $3 \text{ mBq.m}^{-3}$. Using the same principle, acceptable thresholds for the log-Laplace or the log-Cauchy distributions range between $0.1 \text{ mBq.m}^{-3}$ and

$0.5 \text{ mBq.m}^{-3}$.

We should also consider that the log-Cauchy distribution needs a second threshold $j_t$ to be properly defined. Indeed, if for a couple both observation $y_i$ and prediction $(\mathbf{H}\boldsymbol{x})_i$ are equal (usually both equal to zero), then $r$ will naturally tend towards 0 so that the $\mathcal{J}_{\log-\text{Cauchy}}$ tends to $-\infty$ as it can be seen in equation (6) with $j_t$ equal to zero. To prevent that, we can define $j_t = 0.1$ mBq.m$^{-3}$ and

$$\mathcal{J}_{\log-\text{Cauchy}}(\boldsymbol{y}; \mathbf{H}\boldsymbol{x}, r, y_t) = \sum_{i=1}^{N_{\text{obs}}} \ln\left(r + \frac{(\ln(y_i + y_t) - \ln((\mathbf{H}\boldsymbol{x})_i + y_t))^2}{r} + \frac{j_t}{c_{\text{ref}}}\right) \quad (6)$$

with $c_{\text{ref}} = 1 \text{ mBq.m}^{-3}$.

As it will be shown later, the choice of the likelihood has in practice a significant impact on the shape of the posterior distribution. Hence, to better describe the uncertainties of the problem, the approach proposed here is to combine and compare the distributions obtained with these three likelihoods.

## 2.2    Modelling of the errors

The likelihood definition, and therefore the posterior distribution shape, is also greatly impacted by the modelling choice of the error covariance matrix $\mathbf{R}$. The matrix is of size $N_{\text{obs}} \times N_{\text{obs}}$. In real nuclear case studies, the number of observations can be important whereas Bayesian sampling methods can usually estimate efficiently only a limited number of variables. To limit the number of variables, most of the literature (Chow et al., 2008; Delle Monache et al., 2008; Winiarek et al., 2012; Saunier et al.,

2013; Rajaona et al., 2015; Tichý et al., 2016; Liu et al., 2017) describe the error covariance matrix as a diagonal matrix with a unique and common diagonal coefficient. This variance accounts for the observation (built-in sensor noise and bias), models (uncertainty of the meteorological fields and transport model), and representation error between an observation and a modelled





prediction (e.g., Rajaona et al., 2015). Saunier et al. (2019) analyse the impact of such a choice on the same case study of this paper. Indeed, the error is a function of time and space and is obviously not common for every observation-prediction couple.

This critical reduction can lead to paradoxes. With $\mathbf{R} = r\mathbf{I}$, the variance $r$ captures an average of the observation-prediction couples error variances. As seen in appendix A, some observations can tamper the set of measurements and artificially reduce the value of $r$, which prevents the densities of the variables to be well spread. Specifically, let us consider the *non-discriminant* observations. These are observations for which, for any probable source $\boldsymbol{x}$, the observation is almost equal to the prediction. In other words, a non-discriminant observation is an observation which never contributes to discriminate any probable source from an other. It is an observation for which, if $\mathbf{R}$ was modelled as a diagonal matrix with $\mathrm{N}_{\mathrm{obs}}$ independent terms, the variance

$r_i$ corresponding to this observation would be very small or zero. If we model $\mathbf{R}$ as $r\mathbf{I}$, then $r$, capturing an average of the $\mathrm{N}_{\mathrm{obs}}$ variances $r_i$, decreases artificially.

    To deal with the existence of these observations, we propose to use two variances $r_1$ and $r_{\mathrm{nd}}$. The discriminant observations will be associated to the variance $r_1$ while the non-discriminant observations will be associated to the variance $r_{\mathrm{nd}}$ during the

sampling process. Necessarily then, $r_{\mathrm{nd}}$ tends to a very small value.

    In the following, we refer to this algorithm as the observation sorting algorithm. A justification of the use of this clustering using the Akaike information criterion (AIC) is proposed in annex B.

    We now propose a second approach to improve the design of the covariance matrix $\mathbf{R}$ and the estimation of the uncertainties. We propose to cluster observations according to their spatial position in $k$ groups, where observations of the same cluster are

assigned the same observation error variance. This proposal is based on the fact that the modelling part of the observation error is a spatially dependent function. With this clustering, we have $\boldsymbol{x} = (x_1, x_2, \ln \boldsymbol{q}, r_1, ..., r_k)$ the source variables of interest, and $\mathbf{R}$ as a diagonal matrix where the $i$-th diagonal coefficient is assigned a $r_j$ with $j \in \{1, .., k\}$ according to the cluster to which the observation $y_i$ belongs.

    Using both methods, the set of variable $\boldsymbol{x}$ to retrieve becomes $(x_1, x_2, \ln \boldsymbol{q}, r_1, ..., r_k, r_{\mathrm{nd}})$.

## 2.3   Modelling of the meterorology and transport

As explained in section 1.3, the linear observation operator $\mathbf{H}$ is computed with an atmospheric transport model, which takes meteorological fields as inputs. More precisely, the Eulerian ldX model, a part of IRSN's C3X operational platform (Tombette et al., 2014), validated on the Algeciras incident, the ETEX campaign, as well as on the Chernobyl accident (Quélo et al., 2007) and the Fukushima accident (Saunier et al., 2013), is the transport model used to simulate the dispersion of the plume, and

therefore to build the observation operators. To improve the accuracy of the predictions, and therefore reduce the uncertainties, several observation operators computed with various physical parameters configuring the transport model and meteorological fields can be linearly combined. This combination then produces a single prediction forecast, hopefully more skilful than any individual prediction forecast.

    First, ensemble weather forecasts can be used to represent variability into the meteorological fields. The members of the

ensemble are based on a set of $\mathrm{N}_{\mathrm{m}}$ models, where each model can have its own physical formulation. Secondly, considering a meteorology with little rainfall, three parameters of the transport model in particular can have an important impact on the





dispersion of a particle (Girard et al., 2014) and are subject to significant uncertainties: the dry deposition, the height of the release and the value of the vertical turbulent diffusion coefficient (Kz).

Therefore, to create an ensemble of observation operators with both uncertainty in the meteorological fields and in the transport parametrisation, a collection of observation operators $\mathbf{H}_1, ..., \mathbf{H}_{N_e}$ can be computed out of four parameters. More specifically, each $\mathbf{H}_i$ with $i \in \{1, N_e\}$ is the output of a transport model parametrised by

- choosing a member from an ensemble of meteorological fields, hence a discrete value in $[1, N_m]$;

- a constant deposition velocity in $[v_{d,\min} = 0.5.10^{-3}\mathrm{m.s}^{-1}, v_{d,\max} = 5.10^{-3}\mathrm{m.s}^{-1}]$;

- a distribution of the height of the release between two layers defined between 0 and 40 meters, and between 40 and 120
meters, the space being discretised vertically in layers of various heights as described in table 1;

- a multiplicative constant on the Kz values chosen in $[0.333, 3]$;

where each parameter range has been set up based on the work of Girard et al. (2014) and deposition and Kz processes are described in table 1. The ensemble of observation operators is therefore computed from a collection of parameters. This collection is obtained from sampling the values of the parameters inside the intervals.

Once the set of operators has been built, the idea is to combine them linearly to get a more accurate forecast. A weight $w_i$ can be associated with each observation operator of the ensemble:

$$\boldsymbol{y}_S = \mathbf{H}\boldsymbol{x} = \sum_{i=1}^{N_e} w_i \mathbf{H}_i \boldsymbol{x} = \sum_{i=1}^{N_e} w_i \boldsymbol{y}_{S,i} \tag{7}$$

which results in combining linearly the predictions of each member of the ensemble. Each weight $w_i$ is a positive variable to be retrieved. The weights can be included in the set of variables sampled by the MCMC algorithm. They are dependent
on each other through the necessary condition $\sum_{i=1}^{N_e} w_i = 1$ so this means $N_e - 1$ weight variables will be added to $\boldsymbol{x} = (x_1, x_2, \ln \boldsymbol{q}, \mathbf{R}, w_1, ..., w_{N_e-1})$.

Several methods are used to explore reliability, accuracy (level of agreement between forecasts and observations), skill, discrimination (different observed outcomes can be discriminated by the forecasts), resolution (observed outcomes change as the forecast changes), and sharpness (tendency to forecast extreme values) of probabilistic forecasts (Delle Monache et al.,
2006). Rank histograms are used to evaluate the spread of an ensemble. Reliability diagrams (graphs of the observed frequency of an event plotted against the forecast probability of an event) and ROC curves (which plot the false positive rate against the true positive rate using several probability thresholds) are used to measure the ability of the ensemble to discriminate (Anderson, 1996).



## 3 Airborne radioactivity increase in Europe in autumn 2017

### 3.1 Context

In this section, the methods are applied to the detection event of $^{106}$Ru in Europe in autumn 2017. We first provide a brief context for the event, review of earlier studies and describe the observation dataset and the model.

Small quantities of $^{106}$Ru were measured by several European monitoring networks between the end of September and the beginning of October 2017. Inquiries to locate the source, the origin of the $^{106}$Ru being unknown, have been carried out, based on the radionuclide measurements. Correlation methods are used by Kovalets and Romanenko (2017) and Kovalets et al. (2020) to retrieve the location of the source. Saunier et al. (2019) apply deterministic inverse modelling methods to reconstruct the most probable source and release: southern Ural is identified as the most likely geographical location and the total release in the atmosphere is estimated to be 250 TBq. The location, release and errors are also investigated by Dumont Le Brazidec et al. (2020), Tichý et al. (2020) and Western et al. (2020) using Bayesian methods.

The concentration measurements used in this study are available in Masson et al. (2019). The dataset has more than 1000 observations of $^{106}$Ru with detection levels from a few $\mu$Bq.m$^{-3}$ to more than 170 mBq.m$^{-3}$ in Romania. It is described in Fig. 1.

### 3.2 Modelling

#### 3.2.1 Physical parametrisation

All simulations, described in section 3.3, are driven using the ECMWF (European Centre for Medium-Range Weather Forecasts) ERA5 meteorological fields (Hersbach et al., 2020). A single observation operator $\mathbf{H}$ is built with the high-resolution forecast (HRES) reanalysis, in order to study the relevance of the methods presented in sections 2.1 and 2.2. An enhanced ensemble of 50 observation operators is built following the methodology of section 2.3, where the ensemble of meteorological fields is the ERA5 EDA (Ensemble Data Assimilation) of 10 members. As explained in section 2.3, each observation operator of this enhanced ensemble is the output of the transport model based on a random member of the ERA5 EDA and a random physical parametrisation. Table 1 refers to the parameters of the ldX dispersion simulations. The choice of parameters is based on the analysis carried out by Saunier et al. (2019) and Dumont Le Brazidec et al. (2020). As seen in section 2.3 and table 1, the vertical mixing of each member of the enhanced ensemble is a random multiple (between $1/3$ and 3) of the corresponding ECMWF EDA member vertical mixing. Furthermore, the constant dry deposition velocity and the distribution of the height of the release between the two first vertical layers of each member is unique and specific to this member. On most measurement stations, there was no rain event on the passage of the plume except for Scandinavia and Bulgaria. This suggests that wet deposition has a weak influence on the simulations, compared to the other processes.

Simulations are performed from the 22$^{\text{th}}$ of September, 2017 at 00.00 UTC to the 13$^{\text{th}}$ of October, 2017, which corresponds to the time of the last observation. The HRES domain grid $G$ of computation of the operators $\mathbf{H}$ has been chosen based on previous works from the authors (Saunier et al., 2019; Dumont Le Brazidec et al., 2020). The enhanced ensemble domain grid



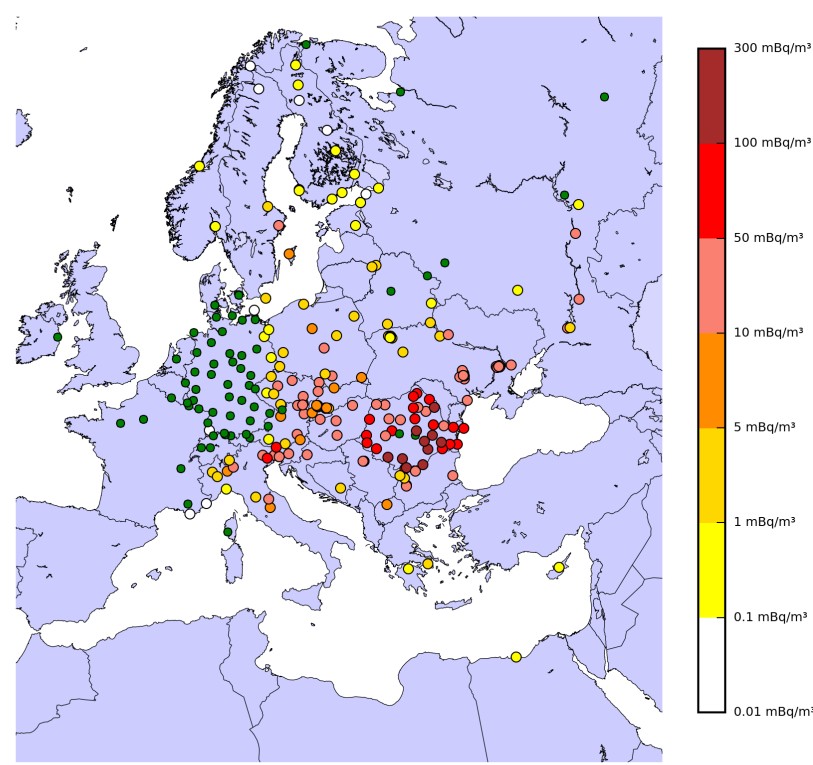

**Figure 1.** Maximum air concentrations of Ruthenium 106 observed over Europe in mBq.m$^{-3}$. Green points measured concentrations below the detection limit.

is of smaller extent due to computation power limitations and focus on the most probable geographical domains of origin of the release, inferred from the HRES results presented in section 3.3.3. The chosen mesh spatial resolution of $G$ is $0.5$ degrees for the simulations with the ECMWF ERA5 HRES observation operator and $1$ degree with the enhanced ensemble of observation operators. The logarithm of the release $\boldsymbol{q}$ is defined as a vector of size $\mathrm{N_{imp}} = 9$ daily release rates from the $22^{\text{th}}$ to the $30^{\text{th}}$ of

September, as explained in (Dumont Le Brazidec et al., 2020).

### 3.2.2 Choice of the priors

In a Bayesian framework, the prior knowledge on the control variables for the $^{106}$Ru source must be described. Following Dumont Le Brazidec et al. (2020), and because no prior information is available, longitude, latitude, observation error, and member weights prior probabilities are assumed to be uniform.





| Parameter | ECMWF ERA5 HRES **H** | Enhanced ensemble |
|---|---|---|
| Computational domain | [6W ; 70E] and [34 N ; 68 N] | [47E ; 62E] and [53 N ; 58 N] |
| Spatial resolution | $0.28125° \times 0.28125°$ | $0.5625° \times 0.5625°$ |
| Vertical resolution | 15 terrain following layers (from 0 to 8000 m) | |
| Time resolution | 1 hour | 3 hours |
| Vertical mixing | K-diffusion following the parametrisation of Louis' closure (Louis, 1979) | Multiple of the HRES vertical |
| | and (Troen and Mahrt, 1986) in unstable conditions in the PBL | mixing depending on the member |
| Horizontal mixing | Constant horizontal eddy diffusion coefficient $K_h = 0 \ \mathrm{m^2 s^{-1}}$ | |
| Wet scavenging | $\Lambda_s = \Lambda_0 p_0$, where $\Lambda_0 = 5.10^{-5} \mathrm{h.(mm.s)^{-1}}$ and $p_0$ is the rainfall intensity in $\mathrm{mm.h^{-1}}$ (Baklanov and Sørensen, 2001) | |
| Dry deposition | constant deposition velocity $v_d = 2.10^{-3} \ \mathrm{m.s^{-1}}$ | $v_d \in [0.5.10^{-3}; 5.10^{-3}] \mathrm{m.s^{-1}}$ |
| Source height | 40% in the first layer (0 to 40m); 60% in the second layer (40 to 120m) | repartition depending on the member |

**Table 1.** Main configuration features of the simulations for the Ruthenium 106 detection event, with the deterministic observation operator or the ensemble of observation operators.

The independent priors of the release rates are chosen as log-gamma distributions to prevent the values of the release to increase unrealistically (Dumont Le Brazidec et al., 2020).

### 3.2.3   Parallel tempering algorithm

We rely on Markov chain Monte Carlo (MCMC) algorithms to sample from the target $p(\boldsymbol{x}|\boldsymbol{y})$. The convergence of a popular MCMC algorithm such as the Metropolis-Hastings (MH) algorithm can be hampered by the encounter of local minima and be
delayed (Dumont Le Brazidec et al., 2020). To overcome this issue, the parallel tempering algorithm (Swendsen and Wang, 1986) is employed. Also called Metropolis-coupled Markov chain Monte Carlo (MCMCMC) or temperature swapping (Earl and Deem, 2005; Baragatti, 2011; Atchadé et al., 2011), it consists in combining several MCMC (such as MH) at different temperatures, where a temperature is a constant flattening out the posterior distribution. Chains with a flat target distribution avoids being trapped in local minima and provide probable source variable vectors to the "real" chain with no temperature.
Details of the implementation applied to our problem can be found in (Dumont Le Brazidec et al., 2020).

### 3.2.4   Parameters of the MCMC algorithm

The transition probabilities used for the random walk of the Markov chains are defined independently for each variable and based on the folded-normal distribution as described by Dumont Le Brazidec et al. (2020). The transition probability of the meteorological member weights is also defined as a folded-normal distribution. Weights are at first updated independently from
each other, and then each proposal is updated by the sum of the weights, i.e.,

$$\forall i \in \{1, \mathrm{N}_e\} \ \ w_i^k \sim \mathcal{FN}(w_i^{k-1}, \sigma_w) \ \text{ then } \ w_i^k = \frac{w_i^k}{\sum_{j=1}^{\mathrm{N}_e} w_j^k} \tag{8}$$





where $\mathcal{FN}$ is the folded-normal distribution, with $\sigma_w$ the prior standard deviation of the weights and $w_i^{k-1}$ the value of the weight of the member $k$ before the walk.

The variances of the transition probabilities are chosen based on experimentations and are set to be $\sigma_{x_1} = \sigma_{x_2} = 0.3$, $\sigma_{\ln q} = 0.03$, $\sigma_r = 0.01$, and $\sigma_w = 0.0005$. All variables are always initialised randomly. Locations of the transitions probabilities are the values of the variables at the current step. When the algorithm to discriminate pertinent observations presented in section 2.2 is used, we consider that a prediction and an observation can be considered as equal if their difference is less than $\epsilon_d = 0.1$ mBq.m$^{-3}$, which is inferred from receptor detection limits described in (Dumont Le Brazidec et al., 2020). Ten chains at temperatures $t_i = c^i$ with $c = 1.5$ are used in the parallel tempering algorithm.

## 3.3  Application of the methods

### 3.3.1  Summary

To see the impacts of the techniques proposed in section 2 (i.e., using diverse likelihoods, new designs of the error covariance matrix and ensemble-based method), the pdfs of the variables describing the $^{106}$Ru source are sampled from various configurations:

– a comparison between the longitude pdf reconstructed with or without the observation sorting algorithm presented in section 2.2 is provided to estimate its efficiency in section 3.3.2 with the enhanced ensemble;

– section 3.3.3 presents results obtained using the HRES meteorology to analyse the impact of using several likelihood distributions. The observation sorting algorithm is applied and two cases are explored: no spatial clustering of the observations, $\boldsymbol{x} = (x_1, x_2, \ln \boldsymbol{q}, r_1, r_{\mathrm{nd}})$ and spatial clustering of the observations with the corresponding modelling of the error covariance matrix $\mathbf{R}$ as described in the end of section 2.2: $\boldsymbol{x} = (x_1, x_2, \ln \boldsymbol{q}, r_1, ..., r_k, r_{\mathrm{nd}})$ where we will use $k = 3$;

– the enhanced ensemble with uncertainty of the dispersion parameters based on the ERA5 EDA of 10 members is analysed in section 3.3.4. Afterwards, pdfs of the source variables $\boldsymbol{x}$ are sampled using the enhanced ensemble: $\boldsymbol{x} = (x_1, x_2, \ln \boldsymbol{q}, r_1, r_{\mathrm{nd}}, w_1, ..., w_{\mathrm{N}_e})$. Results are reconstructed with the help of the observation sorting algorithm and diverse likelihoods.

Note that, when the observation sorting algorithm is used, in all cases, approximately half of the observations are sorted as non-discriminant.

### 3.3.2  Study of the interest of the observation sorting algorithm

We present here an experiment supporting the observation sorting method. A reconstruction of the source variables is proposed using the enhanced ensemble of observation operators, only on the first 30 members for the sake of computation time. The enhanced ensemble is studied later in section 3.3.4. A log-Laplace likelihood with a threshold equal to 0.1 mBq.m$^{-3}$ is used





in two cases: with or without applying the observation sorting algorithm. The source variable vector is therefore $\boldsymbol{x}_{\mathrm{with}} = (x_1, x_2, \ln \boldsymbol{q}, r_1, r_{\mathrm{nd}}, w_1, ..., w_{\mathrm{N}_e})$ (with observation sorting algorithm) or $\boldsymbol{x}_{\mathrm{without}} = (x_1, x_2, \ln \boldsymbol{q}, r, w_1, ..., w_{\mathrm{N}_e})$ (without).

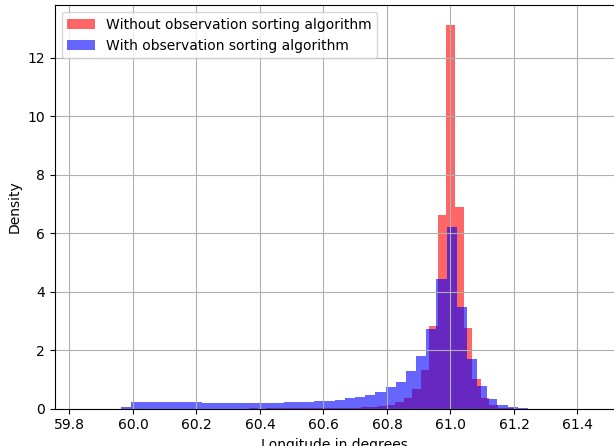

**Figure 2.** Density of the longitude for a log-Laplace likelihood with threshold $0.1$ mBq.m$^{-3}$ of the Ruthenium source sampled with the enhanced ensemble of observation operators using the parallel tempering method with or without the help of the observation sorting algorithm.

Figure 2 represents the longitude variable pdf of the sources variables vectors. The blue histogram, which represents the
longitude pdf in the case where the observation sorting algorithm is applied, is far more spread out than the case without. It indeed ranges from 60 to 61.2 degrees while the red longitude density (without applying the observation sorting algorithm) ranges from 60.8 to 61.2, i.e., the pdf extent with sorting is the triple of the spread without.

The mean of the observation error variance $r$ samples in the case without the observation sorting algorithm is $0.399$. In the case with the observation sorting algorithm, the discriminant observation error variance $r_1$ is equal to $0.63$ and the non-
discriminant observation error variance $r_{\mathrm{nd}}$ tends to a very low value. We note $\boldsymbol{x}_{\mathrm{with}}$ the set of sources sampled (and therefore considered as probable) when using the observation sorting algorithm and $\boldsymbol{x}_{\mathrm{without}}$ the set of sources sampled without. For example, a source $\boldsymbol{x}$ in $\boldsymbol{x}_{\mathrm{with}}$ is of longitude $x_1$ in $[60, 61.2]$. By definition of the log-Laplace pdf in equation (5), the fact that $r_{\mathrm{nd}}$ tends to a very low value confirms that non-discriminant observations are observations which are not able to discriminate between any of the sources in $\boldsymbol{x}_{\mathrm{with}}$.

What happens when the observation sorting algorithm is not used (red histogram), i.e., with the basic design $\mathbf{R} = r\mathbf{I}$, is that the observation error variance $r$ is common for all observations. The resulting $r = 0.399$ is therefore a compromise between $r_1 = 0.63$ and $r_{\mathrm{nd}} \sim 0$ the variances sampled with the sorting algorithm. In other words, the uncertainty on the discriminant observations is reduced compared to the uncertainty found when applying the sorting algorithm. That is, the confidence in the discriminant observations is artificially high. This is why the extent of the resulting posterior pdfs is reduced comparing to
the case with the observation sorting algorithm. More precisely, the probability of the most probables sources (here longitudes





between 60.8 and 61.2) is increased and the probability of the least probable sources is decreased (longitudes between 60 and 60.8).

The observation sorting algorithm is a clustering algorithm that avoids this harmful compromise. Observations with very high likelihood (i.e., very small observation error variance, very high confidence for all probable sources) - the non discriminant
observations - are assigned a specific observation error variable. In this way, the uncertainty variance associated to the other observations is far more appropriate. This clustering is totally legitimate as explained in annex B.

Finally, note that sampling the longitude posterior distribution using $\boldsymbol{y}_\mathrm{d}$ the set of discriminant observations instead of $\boldsymbol{y}$ yields a pdf very similar to the blue density in Fig. 2. In other words, without the observation sorting algorithm application, it means that considering observations which cannot discriminate between a source of longitude 60 and a source of longitude
60.8 actually decrease the probability of the source of longitude 60 which is a non-sense. The significant difference between the two pdfs makes the application of the algorithm necessary.

### 3.3.3 Sampling with the HRES data, several likelihoods, the observation sorting algorithm, and with or without observation spatial clusters

In this section, we study two cases. First, we assess the impact of the choice of the likelihood on the reconstruction of the
control variable pdfs ($\boldsymbol{x} = (x_1, x_2, \ln \boldsymbol{q}, r_1, r_\mathrm{nd})$) and secondly, we investigate how assigning diverse error variance terms to the observations according to a spatial clustering can affect the results ($\boldsymbol{x} = (x_1, x_2, \ln \boldsymbol{q}, r_1, r_2, r_3, r_\mathrm{nd})$). In this second case, clusters are computed with a k-means algorithm for $k = 3$. Observations clusters are presented in the map below. As it can be seen in Fig. 3, the first cluster gathers western and central Europe observations with a very low error variance, which indicates that predictions of the model are a good fit to the observations. A second cluster corresponds to the eastern Europe observations
with a large variance ($r = 0.64$), which is consistent with the fact that most of the important measures belong in this cluster. And finally, the third cluster corresponds to Russia with $r = 0.24$. We now present the reconstruction of the pdfs in the two scenarios.

Figures 4.a, 4.c, 5.a show the marginal pdfs of the variables describing the source using the observation sorting algorithm and for several likelihoods, using the HRES meteorology. The longitude pdf support ranges from 58.3 to almost 61.5 degrees, and
the latitude support from 55 to 57 degrees. The extent of the joined longitude pdfs is far greater than the extent of any longitude pdf reconstructed using any likelihood distribution. This shows that using a single likelihood is not enough to aggregate the whole uncertainty of the problem. We can see on these graphs that the choice of the likelihood has a greater impact than the threshold choice on the final coordinates pdfs. Nevertheless, they are in general all consistent in the pointed area of Ruthenium release, especially given that the observation operators interpolation step is 0.5 degrees.
The daily Total Retrieved Released Activity (TRRA) was mostly significant on the 25[th] of September. The extent of the release pdfs overlap is bigger than the coordinates pdfs overlap extent; probable TRRA values range from 60 to 200-250 TBq. However, the daily TRRA pdfs obtained from the log-normal and the log-Laplace choices are significantly impacted by the threshold value choice.



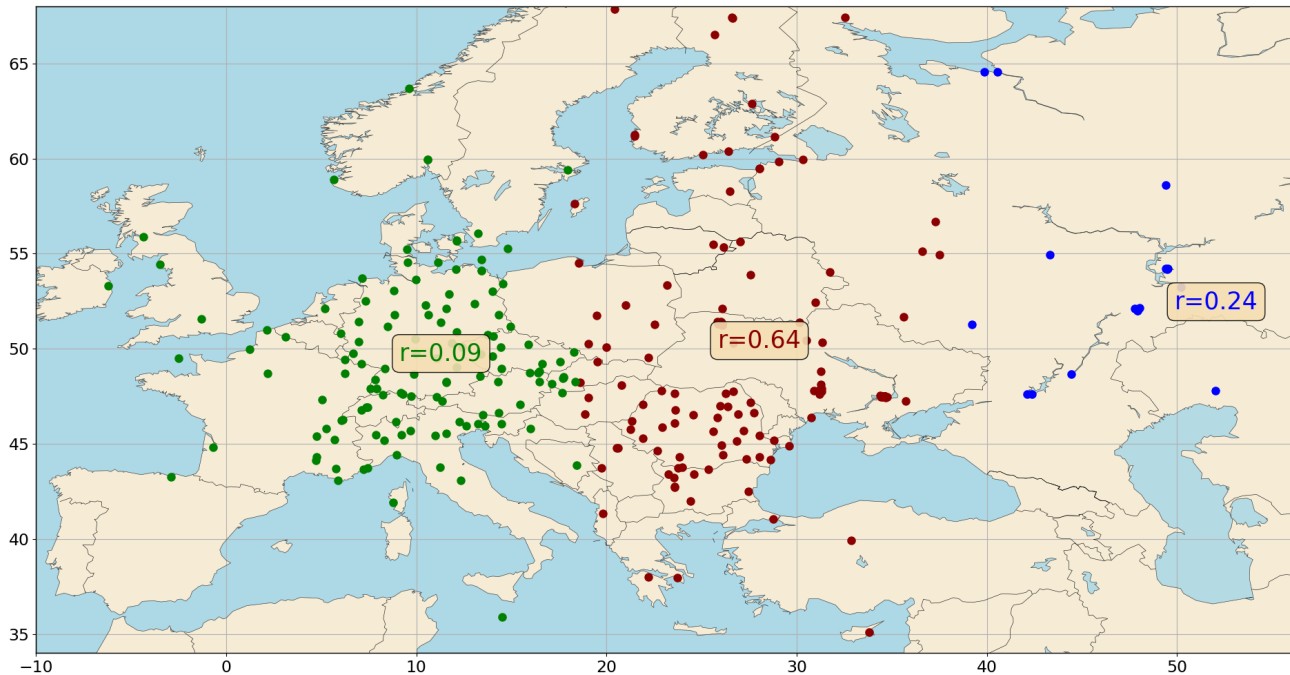

**Figure 3.** $^{106}$Ru observations spatially clustered with a k-means algorithm ($k = 3$). The means of the corresponding three variance distributions have been computed using a parallel tempering algorithm for a log-normal distribution with a threshold equal to 0.5 mBq.m$^{-3}$.

Figures 4.b, 4.d, 5.b show the marginal pdfs of the variables describing the source in the same configuration, except that
three error variances are used and assigned to the observations using the spatial clustering. The results are impacted by this
clustering. More precisely, the pdfs of the coordinates of the various likelihoods are more alike and most of them are centred
around the coordinates $[60.5, 55.8]$, 30km away from the Mayak Nuclear Facility. The TRRA has increased from $[80, 220]$ TBq
in Fig. 5.a to $[100, 270]$ TBq in Fig. 5.b.

### 3.3.4   Sampling from the enhanced ensemble with weights interpolation

Before reconstructing the pdfs of the Ruthenium source using the observation operators ensemble, we study the dispersion of
this enhanced ensemble, created by sampling on the transport model parameters and the ERA5 EDA. A number of 50 members
are used to create the enhanced ensemble.

The original ERA5 EDA meteorology is under-dispersive as it can be seen in Fig. 6a: in blue is drawn the HRES meridional
wind predictions, and in red the EDA meridional wind mean prediction with the maximum and minimum values, for a random
location in Europe. For most of the times, the ensemble values do not even recover the HRES value, which indicates that
the ensemble has a small dispersion. ECMWF ensemble forecasts indeed tend to be under-dispersive in the boundary layer
(especially in the short range) (Pinson and Hagedorn, 2011; Leadbetter et al., 2020). Furthermore, the release height parameter



**Figure 4.** Pdfs of the coordinates describing the Ruthenium source sampled using the parallel tempering method and the observation sorting algorithm and for various likelihoods in two scenarios: Longitude without (a) and with observation spatial clustering in three clusters (c) and Latitude without (b) and with (d). L-L means log-Laplace, L-n means log-normal, L-C means log-Cauchy and $y_t$ is the likelihood threshold.





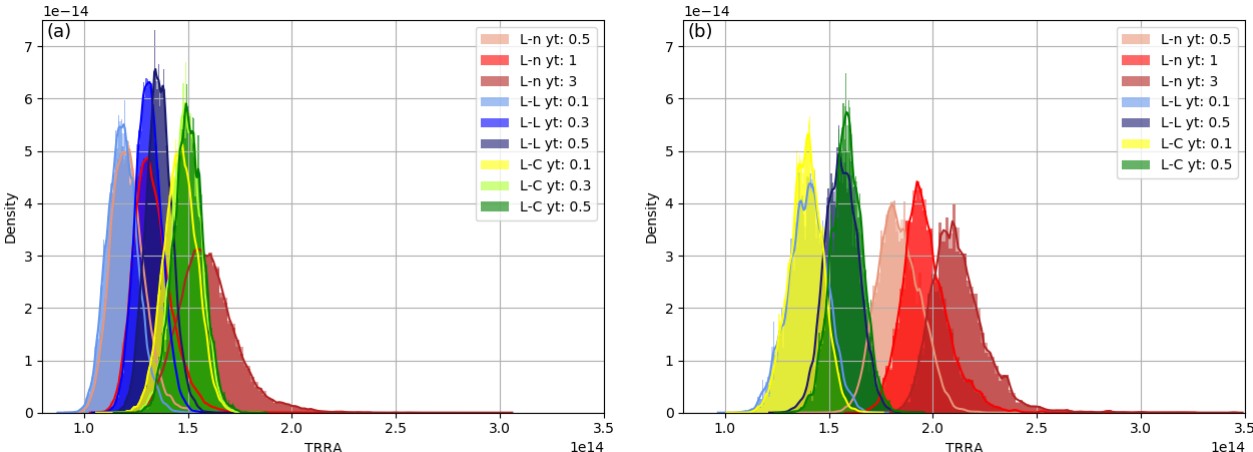

**Figure 5.** Pdfs of the Total Retrieved Released Activity (TRRA) describing the Ruthenium source sampled using the parallel tempering method and the observation sorting algorithm and for various likelihoods in two scenarios: without (a) and with observation spatial clustering in three clusters (b). L-L means log-Laplace, L-n means log-normal, L-C means log-Cauchy and $y_t$ is the likelihood threshold.

has low chances to be of significant impact. Therefore, with only two variables (Kz and dry deposition velocity) with high chances to be of significant impact to sample, we consider that 50 members is an adequate number.

To examine the spread of the ensemble of observation operators, we need to define a reference source $x_{\mathrm{ref}}$ for which predictions of the ensemble will be computed for each member and compared afterwards with the $^{106}$Ru observations. Note that the reference source choice is necessarily an arbitrary choice which biases the results. We use the most probable source from Saunier et al. (2019) as reference source: the reference source is located in $[60, 55]$ and with a release of 90 TBq on the 25$^{\mathrm{th}}$ of September and of 166 TBq on the 26$^{\mathrm{th}}$ of September. The corresponding rank diagram (Fig. 6b) shows that the

predictions often underestimate the observation value. A ROC and a reliability diagram are provided in annex C to study more deeply the ensemble of observation operators.

  We now study the impact of adding meteorological and transport uncertainties into the sampling process:

  $x = (x_1, x_2, \ln q, r_1, r_{\mathrm{nd}}, w_1, ..., w_{\mathrm{N}_e})$. The integration of an enhanced ensemble to deal with the meteorological and dispersion uncertainties has a very interesting impact on the reconstruction of the TRRA. In Fig. 7d, the TRRA ranges from 100-150

to 300-350 TBq, and the standard deviation (std) of the joint multi-model TRRA is therefore far more important than the std of the joint HRES TRRA. Note also that with the log-normal likelihood and a threshold of 0.5 mBq.m$^{-3}$, 1 mBq.m$^{-3}$ or 3 mBq.m$^{-3}$, or log-Laplace with a threshold of 0.5 mBq.m$^{-3}$, the release is split between the 25$^{\mathrm{th}}$ and the 26$^{\mathrm{th}}$ as it can be noted in Fig. 7e and 7f. In other words, the integration of weights member interpolation adds uncertainty not only over the magnitude of the release but also over the timing of the release (here, the day). However, pdfs of the longitude and latitude are

not significantly impacted as it can be seen in Fig. 7a and 7b.

  The pdfs of the member weights are displayed in Fig. 8 for several likelihoods and thresholds. Only weights pdfs with high medians are included in the graphs for reasons of visibility. Several remarks are in order: member 27 is always included in the





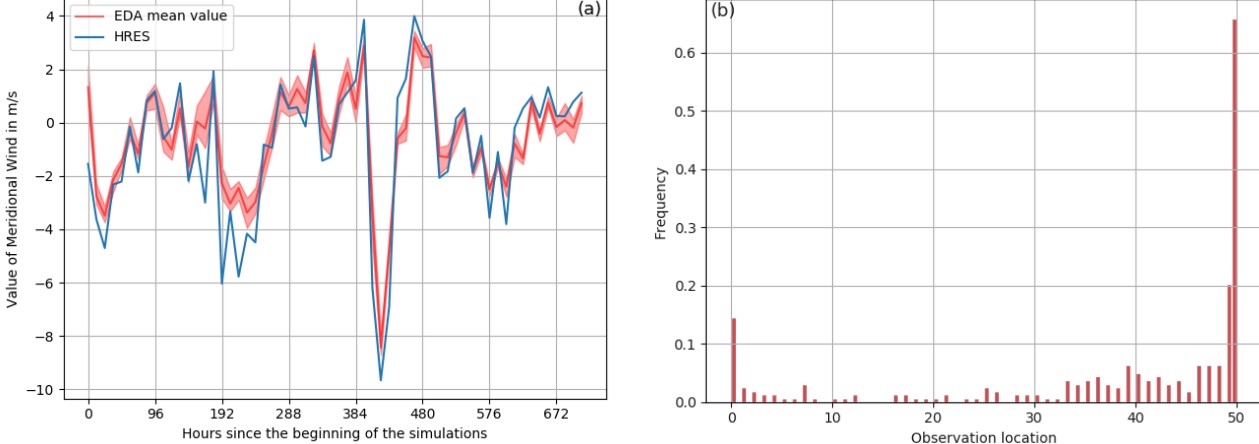

**Figure 6.** Evolution of the meridional wind of the HRES and EDA meteorologies for a random location in Europe since the beginning of the simulations. The red curves represents the mean of the meridional wind EDA members with the space between the minimum and maximum values (a). Rank histogram: comparison between the $^{106}$Ru observations and the reference predictions of the enhanced ensemble (b).

mix of weights which define the interpolated observation operator used to make the predictions and is often one or the most important part of this mix. Since the ERA5 EDA is not very dispersive, we can make the hypothesis that the weight density

of a member mainly depends on the dispersion parameters which are used for creating this member (or observation operator). The height layer of the release for the operator member 27 is mainly the one between 40 and 120 meters (>90%), its deposition constant velocity is $0.6 \times 10^{-3} \mathrm{m.s}^{-1}$ which is very close to the lower bound (minimal possible deposit) and the Kz has been multiplied by $0.47$. It also corresponds to member 6 of the ERA5 EDA.

Member 44 is present 5 times and corresponds to a deposition velocity of $1.2 \times 10^{-3} \mathrm{m.s}^{-1}$, a release mainly on the second

layer (80%) and a Kz multiplied by $0.34$. Member 42 is present 4 times and corresponds to a deposition velocity of $2.1 \times 10^{-3} \mathrm{m.s}^{-1}$, a release also mainly on the second layer (83%) and a Kz multiplied by $0.41$. Member 47 is present 2 times and corresponds to a deposition velocity of $0.92 \times 10^{-3} \mathrm{m.s}^{-1}$, a release split between the two layers and a Kz multiplied by $1.1$. Two hypotheses can be made: the weight of member 27 (and member 47) is large because it has a very small deposition velocity and that deposition velocity is overestimated when using the standard choice. Secondly, Kz may well be overestimated, and

the release should be on the second layer (between 40 and 120m) since these are the two common points between the three more relevant members (27,44, and 42). Note that a first try with 30 members (instead of 50 members here) led to the same conclusions (except for the height of the release on which nothing could be concluded).

### 3.4 Summary and conclusions

In this paper, we proposed several methods to quantify the uncertainties in nuclear atmospheric source inverse modelling.

We first investigated the choice of the likelihood. We described with both a theoretical discussion and an application why





**Figure 7.** Pdfs of the variables describing the Ruthenium source sampled using the parallel tempering method and the enhanced ensemble of observation operators and the observation sorting algorithm and for various likelihoods: longitude (a), latitude (b), TRRA on the main day (c), TRRA (d), TRRA on the 25$^{\text{th}}$ of September (e) and TRRA on the 26$^{\text{th}}$ of September (f). TRRA pdfs are drawn using log-scale y-axis. L-L means log-Laplace, L-n means log-normal, L-C means log-Cauchy and $y_t$ is the likelihood threshold.





**Figure 8.** Densities of the weights of the members of the enhanced ensemble using the parallel tempering method and the observation sorting algorithm for diverse likelihoods: log-normal with threshold 0.5 (a), log-normal with threshold 3 (b), log-Laplace with threshold 0.1 (c), log-Laplace with threshold 0.5 (d), log-Cauchy with threshold 0.3 (e), log-Cauchy with threshold 0.5 (f). L-L means log-Laplace, L-n means log-normal, L-C means log-Cauchy and $y_t$ is the likelihood threshold. Only densities with high medians are shown.



it can have a major impact on the final probability density function shape. We also discussed the need to better design the error covariance matrix to avoid observations with low discrimination power to artificially decrease the spread of the posterior distributions. Moreover, we provided a method to add meteorological and dispersion uncertainties to the reconstruction of the distributions of a source, improving its evaluation. Following a Bayesian approach, each member of the enhanced ensemble is

given a weight which is sampled in the MCMC algorithm.

A full reconstruction of the variables describing the source of the Ruthenium 106 in September 2017 and their uncertainty was provided. The three methods have proven to be useful. In particular, the choice of the likelihood has shown to be of great influence on the release coordinates distribution. On the other hand, the addition of weights as variables to be sampled had a major impact on the shape of the magnitude and time of the release pdf.

We intend to apply the methods developed in this paper to the release of the Fukushima-Daiichi accident.

**Appendix A: False paradox of the discriminant and non-discriminant observations**

Let us suppose that the cost function is computed from a Gaussian likelihood, then we have

$$\mathcal{J}(\boldsymbol{x}|\boldsymbol{y}) = \frac{1}{2}\sum_{i=1}^{\mathrm{N_{obs}}} \frac{(y_i - (\mathbf{H}\boldsymbol{x})_i)^2}{r} + \frac{\mathrm{N_{obs}}\ln(r)}{2}. \tag{A1}$$

Suppose we add to this set of observations $\boldsymbol{y}$ of size $\mathrm{N_{obs}}$ a second set of observations $\boldsymbol{y}^* = \mathbf{0}_{\mathrm{N^*_{obs}}}$ of size $\mathrm{N^*_{obs}}$ for which the

corresponding predictions are also zero. This can happen for instance if we add observations preceding the accident.

If we take into account this new set of observations, the cost function becomes:

$$\begin{aligned}
\mathcal{J}(\boldsymbol{x}|\boldsymbol{y}, \boldsymbol{y}^*) &= \frac{1}{2}\sum_{i=1}^{\mathrm{N_{obs}}+\mathrm{N^*_{obs}}} \frac{(y_i - (\mathbf{H}\boldsymbol{x})_i)^2}{r} + \frac{1}{2}(\mathrm{N_{obs}} + \mathrm{N^*_{obs}})\ln(r) \\
&= \frac{1}{2}\sum_{i=1}^{\mathrm{N_{obs}}} \frac{(y_i - (\mathbf{H}\boldsymbol{x})_i)^2}{r} + \frac{1}{2}(\mathrm{N_{obs}} + \mathrm{N^*_{obs}})\ln(r)
\end{aligned} \tag{A2}$$

since for each observation of the new set, $y_i = (\mathbf{H}\boldsymbol{x})_i = 0$. Suppose now that $\mathrm{N^*_{obs}}$ goes to infinity, then the observation error variance $r$ should tend to 0. As $r \to 0$, the distributions sampled are being more and more peaked.

Therefore in this configuration, adding a given number of observations anterior to the accident will degrade the distributions of the source variables. This problem is due to the homogeneous and hence inconsistent design of the observation error covariance matrix. Assigning a different $r$ to this new set of observations would solve effectively this false paradox.





## Appendix B: Study of the observation sorting algorithm clustering with the Akaike Information Criterion (AIC)

We compare a first model (0) where only one variable $r$ is used to describe the covariance matrix $\mathbf{R}$ and a second model (d)

where two variables $r_1$ et $r_2$ are describing $\mathbf{R}$ with the AIC (Hastie et al., 2009). Therefore

$$\text{AIC}(0) = 2 - 2\ln(L(0)); \tag{B1}$$

$$\text{AIC}(d) = 4 - 2\ln(L(d)) \tag{B2}$$

where $L(0)$ and $L(d)$ are the maximum likelihoods when using the first and second model respectively. Using Gaussian likelihoods to facilitate calculations and out of normalisation contants:

$$\frac{1}{2}\text{AIC}(0) = 1 + \frac{N}{2}\ln\frac{S}{N} + \frac{N}{2} \tag{B3}$$

$$\frac{1}{2}\text{AIC}(d) = 2 + \frac{N_1}{2}\ln(r_1) + \frac{S_1}{2r_1} + \frac{N_2}{2}\ln(r_2) + \frac{S_2}{2r_2} \tag{B4}$$

where $S_k = \sum_{i=1}^{N_k}(y_i - (\mathbf{H}\boldsymbol{x})_i)^2$ and $N_1$, $N_2$ the number of observations assigned to the first and second model, respectively ($N_1 + N_2 = \text{N}_{\text{obs}}$). Therefore, comparing the max likelihoods of the two models:

$$\frac{1}{2}(\text{AIC}(0) - \text{AIC}(d)) = -1 + \frac{N_1}{2}\ln\frac{(S_1 + S_2)N_1}{(N_1 + N_2)S_1} + \frac{N_2}{2}\ln\frac{(S_1 + S_2)N_2}{(N_1 + N_2)S_2} \tag{B5}$$

We can note that the smaller $S_2$ (and therefore the bigger $S_1$), the better the model (d) comparing to (0). The observation sorting algorithm exactly aims at selecting a large set of observations (the non-discriminant observations $\boldsymbol{y}_{\text{nd}}$) with a very small maximum likelihood $S_2 = S_{\text{nd}}$.

We write $R = \frac{S_1}{S_2}$ and $M = \frac{N_1}{N_2}$ and use $N_1 + N_2 = \text{N}_{\text{obs}}$, then:

$$\frac{1}{2}(\text{AIC}(0) - \text{AIC}(d)) = -1 + \frac{\text{N}_{\text{obs}}M}{2(1+M)}\ln\left(\frac{1+\frac{1}{R}}{1+\frac{1}{M}}\right) + \frac{\text{N}_{\text{obs}}}{2(1+M)}\ln\left(\frac{1+R}{1+M}\right) \tag{B6}$$

and we can draw AIC(0) - AIC(d).

According to the AIC criterion, the model (d) with two variables is judged useless if the average likelihood of the observations $y_1$ linked to group 1 is close to the average likelihood of the observations $y_2$ linked to group 2. The observation sorting algorithm exactly aims at creating two groups of observations: one where the average likelihood of the observations is close to 0 and another one where the average likelihood of the observations is high. In other words, the ratio $(S_1/S_2)$ tends towards

infinity (depending on the choice of $\epsilon_d$) and the ratio $(N_1/N_2)$, for example, is equal to 1. That is, the sorting algorithm creates two groups such that the corresponding coordinates $(S_1/S_2)$ and $(N_1/N_2)$ in Fig. B1 are as far away from negative values as possible. Therefore, the AIC criterion totally justifies the need of the clustering accomplished by the observation sorting algorithm.

## Appendix C: ROC and reliability diagram of the observation operators ensemble

A ROC and a reliability diagram are computed using the reference source defined in section 3.3.4 to assess the ability of the forecast to discriminate between events and non-events and its reliability, respectively (Delle Monache et al., 2006). A good

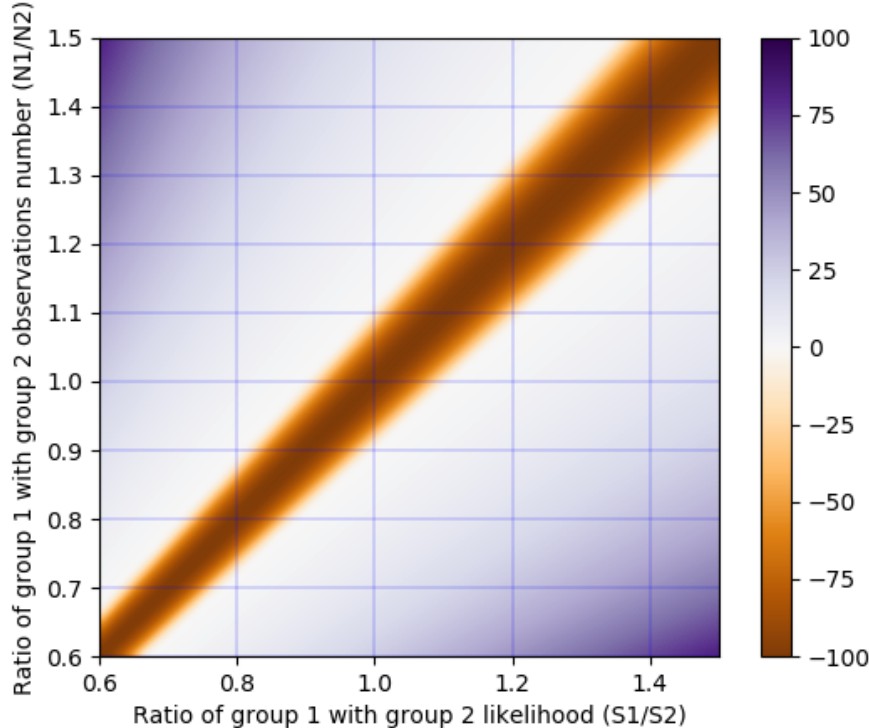

**Figure B1.** AIC(0)-AIC(d) with negatives values multiplied by 100 to be more visible. Negative values indicate the cases where the model (0) with one variable is preferable. Positives values indicate the cases where the model (d) with two variables is preferable. In x-axis is the ratio between the likelihoods $S_1$ and $S_2$ linked to variables 1 and 2, respectively, of the model (d). In y-axis is the ratio between the numbers of observations $N_1$ and $N_2$ linked to variables 1 and 2, respectively, of the model (d).

ROC curve is as close as possible to $1$ in probability of a hit and to $0$ in probability of a false occurence. We recall that a hit for a certain threshold $t$ is when the observation belongs to the considered interval and a number of corresponding predictions greater than the threshold $t$ belong to the considered interval. A false occurence is when the observation does not belong to the
considered interval but a number of corresponding predictions greater than the threshold $t$ belong to the considered interval. The ROC is plotted for a list of thresholds $t$.

Each curve of Fig. C1a and C1b corresponds to a dichotomous event: $y \in [y_{\min}; y_{\max}]$ where $y$ is an observation, and $y_{\min}, y_{\max}$ the values that define whether the event is true or false for $y$. These indicators are plotted for several ranges $[y_{\min}, y_{\max}]$. A reliable ensemble, for a given event, has a reliability curve as close as possible to the identity function.
From the ROC curves, the enhanced ensemble appears to be good for discriminating: curves always have a low rate of false occurrence and an acceptable hit rate. In the reliability diagrams, the forecast overestimates the probability that an observation



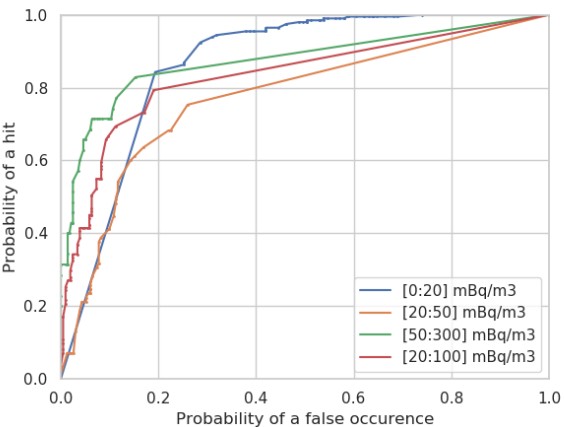
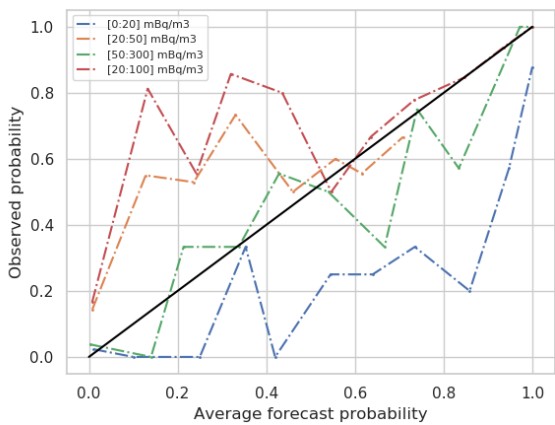

**Figure C1.** ROC curve and reliability diagram of the enhanced ensemble created with sampling of EDA and the transport model parameters.

is between 0 and 20 mBq.m$^{-3}$ which relates to predictions underestimating the observations in general. For the three other events, the diagrams show an acceptable reliability in the enhanced forecast.

*Data availability.* The observation dataset used is described in detail and is publicly available in the work of Masson et al. (2019) as
supplementary information.

*Author contributions.* Joffrey Dumont Le Brazidec: Software, Methodology, Conceptualization, Investigation, Writing - Original Draft, Visualization. Marc Bocquet: Methodology, Conceptualization, Writing - Review and Editing, Visualization, Supervision. Olivier Saunier: Resources, Methodology, Conceptualization, Writing - Review and Editing, Visualization Supervision. Yelva Roustan: Methodology, Software, Writing - Review and Editing

*Competing interests.* The authors declare that they have no conflict of interest.

*Acknowledgements.* The authors are grateful to the European Centre for Medium-Range Weather Forecasts (ECMWF) for the meteorological ERA5 fields used in this study. CEREA is a member of Institut Pierre Simon Laplace (IPSL). They also wish to thank Didier Lucor, Yann Richet and Anne Mathieu for their comments on this work.



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
