# Peer review of "Quantification of uncertainties in the assessment of an atmospheric release source applied to the autumn 2017 $^{106}\text{Ru}$ event"

_Atmospheric Chemistry and Physics, 2020_

## Author Comment (AC2)

**Discussion: Quantification of uncertainties in the assessment of atmospheric release source with application to the autumn 2017 106Ru event**

Joffrey Dumont Le Brazidec1,2, Marc Bocquet2, Olivier Saunier1, and Yelva Roustan2 1IRSN, PSE-SANTE, SESUC, BMCA, Fontenay-aux-Roses, France 2CEREA, Joint laboratory École des Ponts ParisTech and EDF R&D, Université Paris-Est, Marne-la-Vallée, France **Correspondence:** Joffrey Dumont Le Brazidec (joffrey.dumont@enpc.fr)

**Report 2**

We thank the reviewer for his/her constructive comments, revisions and technical comments, which allowed us to clarify some points of the paper.

The manuscript presents an evaluation of the impact on assumptions surrounding statistical model selection on posterior estimates of source properties of (unknown) radiological releases. This manuscript will be informative to researchers and operational users. I have tried to avoid repetition if the previously posted comment. One major drawback of the manuscript is that much of the motivation is based on a straw-man argument, i.e. the original Gaussian set-up is designed in the manuscript such that it will fail. Below are a number of suggestions for revisions to improve the manuscript, followed by technical comments.

– Paragraph starting line 110: This is a straw-man argument. The assumption from the start is that the error is larger for higher measurements. This may not always be valid, e.g. an incorrectly dispersed wide plume with a very high concentration. It may be that the error is much larger for the smaller measurements than the larger measurements. There are other approaches to improve the validity of Gaussian (or any) likelihoods through transformation of variables. For example, using a non-linear forward model. Caveats, justification and 'typical errors' needs explaining, preferably at the start of section 2.

We agree that the fourth criterion results from a "multiplicative" consideration of the error distribution, which is not necessarily true. However, we believe that this simplification is relevant in the case where a limited number of hyperparameters are available to model  $\mathbf{R}$ . We have specified in the text that this criterion relies on this underlying simplification.

- Much of the arguments surround having independent and identically distributed (iid) model-measurement error in the covariance. Many of the arguments throughout can be countered by the use of non-iid covariances, e.g. tR, where the diagonal of R is the measurement value and t is a scaling – equivalent to having e.g. a 10% model-measurement error. This needs further discussions and better justification for the arguments proposed (e.g. non-negativity).

This is true, our argument is indeed based on the hypothesis of modelling **R** by some hyperparameters. However, it should be noted that the variance hyperparameters of the matrix **R** are estimated through a hierarchical Bayesian approach inside the MCMC procedure. In other words, it is not estimated independently of the MCMC in an empirical Bayesian approach. More precisely, if we note  $\mathbf{x}_S = (x_1, x_2, \ln \mathbf{q})$  the vector of variables describing the source except for the hyperparameter r, we try to estimate :

$$p(\boldsymbol{x}_{S}, r_{1}, ..., r_{N} | \boldsymbol{y}) \propto p(\boldsymbol{y} | \boldsymbol{x}_{S}, r_{1}, ..., r_{N}) p(\boldsymbol{x}_{S}, r_{1}, ..., r_{N})$$

$$\propto p(\boldsymbol{y} | \boldsymbol{x}_{S}, r_{1}, ..., r_{N}) p(\boldsymbol{x}_{S} | r_{1}, ..., r_{N}) p(r_{1}, ..., r_{N})$$
(1)

and

$$p(\boldsymbol{x}_{S}, r_{1}, ..., r_{N} | \boldsymbol{y}) = p(\boldsymbol{x}_{S} | \boldsymbol{y}, r_{1}, ..., r_{N}) p(r_{1}, ..., r_{N} | \boldsymbol{y})$$
(2)

If we model  $\mathbf{R}$  with a term t of scaling model-measurement, we suppose that the values  $r_1, ..., r_N$  depend on t, i.e., we suppose that these values depend on the difference between the observations  $\boldsymbol{y}$  and the predictions  $\boldsymbol{y}_S = \mathbf{H}\boldsymbol{x}_S$ . However, this difference depends on the definition of  $\boldsymbol{x}_S$ . In other words, with a definition of  $\mathbf{R}$  depending on t then  $r_1, ..., r_N$  depend on  $\boldsymbol{x}_S$ . This would not be rigourously defined in a hierarchical Bayesian formalism. Furthermore, even if we agree that  $\mathbf{R} = r\mathbf{I}$  is a simplification and that better and more rigorous modelling are possible,

this is a very classic choice in the literature.

Section 3.2.3: It would be useful for many readers to provide a brief conceptual introduction to MCMC methods (i.e. asymptotically exact methods not reliant on closed form solutions or conjugacy).

A few sentences have been added.

- Section 3.4: This section needs expanding considerably. It is a paper with lot of content. At current, the summary provides an overview of the approach of the paper but no summary of the finding. The summary should summarise the results and findings to adequately inform the lazy reader on the paper's content.

This is true and we have extended the conclusion accordingly. Thank you.

- I understand the following would require a lot of extra work, and so do not mandate it for publication. It would however, much improve the paper. Seeing as an ensemble is used, it would seem sensible to me to use a simulated dataset (a simulation using an ensemble member) to draw conclusions from the various experiment. It is not perfect, but would be useful to have a "truth".

Indeed, we agree that this may be interesting to use a synthetic case. However, we decided in this paper to focus on the development of methods rather than on the maximum exploitation of the results on the  $^{106}$ Ru case.

 It would be useful in the analysis and plot to also show the original case of a Gaussian likelihood. This is needed to prove the worth of using a non-Gaussian likelihood. We believe that the relevance of a non-Gaussian likelihood can be proved in theory but not on a particular case (Liu et al., 2017). Inverse modelling with a Gaussian likelihood might provide pertinent results in some cases : the problem is that these results are based on the use of the few largest observations instead of the whole dataset. The information in some of the data is not exploited, which can lead to discrepancies between the reconstructions in some cases. The longitude and latitude distributions are reconstructed with a Gaussian likelihood in figure 1. We can note that the results differ significatively from the ones obtained using log-normal, log-Laplace, or log-Cauchy likelihoods.

**Figure 1.** Density of the coordinates describing the  ${}^{106}$ Ru source for a Gaussian likelihood of the Ruthenium source sampled using the parallel tempering method with the help of the observation sorting algorithm and the HRES meteorology.

Technical comments:

Title: "modelling uncertainties" is ambiguous as it can refer to a statistical model or a transport model. The title also is
not grammatically correct. A suggested improvement is "Quantification of uncertainties in atmospheric release source
assessment applied to the autumn 2017 Ruthenium 106 source".

We have changed the title following the suggestion of the reviewer. Thank you.

Abstract, line 5: "improve on these distributions" is vague. 'Better quantify' or 'improve estimates of these distributions' would be better.

We have corrected the sentence following this suggestion.

- Line 7: A space is not needed before a colon in English.

This has been corrected. Thank you.

- Line 8: Clarify 'model errors' (I assume transport errors?)
   We have added « physical » to clarify.
- Line 10: 'several suited distributions for the errors are advised' the passive voice reads as though you are advising the distributions. Better "we suggest several suitable distributions for the errors" or "are suggested" if sticking with the passive.

Active voice in now used. Thank you for this advice.

- Line 17: sources or a source; remove 'many'; 'Therefore' doesn't follow delete.
   This has been corrected, Thank you for these corrections.
- Line 31: 'finds it origin in' to 'originates from'

This has been corrected. Thank you.

 Line 38: This is an oversimplification of the weighting strategy. See, for example, importance resampling or Ensemble Kalman filter methods.

We use a linear weighting strategy in order to be able to include the weights as variables of the ensemble to be sampled.

- Line 46: Why index vector x elements with x1, x2 and then ln(q)? Perhaps x3=ln(q) would be clearer.

Classic coordinates names are (x,y) but y name is already used for describing the observations vector.  $\ln q$  being a vector, this might be confusing to use  $x_3$ .

- Line 47: R is better described as the covariance matrix containing the model-measurement errors.

This has been corrected.

Line 52: Introduce for the reader what the prior is (i.e. the probability distribution of prior knowledge before considering data).

This has been added.

- Line 57: reconstructed posterior distribution

This has been corrected. Thank you

- Line 60: 'transformation' is better than 'parameterisation'

This has been corrected.

 Line 61: 'are the results' would be better as 'are the results of a simulation' This has been corrected.

- Line 62: 'and are therefore depending' to 'and depend on' This has been corrected. Thank you a lot.
- Line 70: This isn't an expansion.

We mean that the computation time is high.

- Line 103: A cost function is a non-probabilistic concept and so better to refer to as simply the negative log-likelihood.

The authors do agree but the term of "cost" or "loss" is actually widespread in the statistic, data assimilation, or inverse problem literature. The use of the term "cost" rather than "negative log-likelihood" allows us to save a lot of space considering the number of occurrences of the term. Furthermore, the cost can correspond to the negative of the log-posterior distribution.

- Equation 4: There should be no divide by 2 in the first term.

This comment has been removed by the reviewer.

- Line 112: A space not full stop is needed between units
   These have been corrected. Thank you.
- Line 115: Capital G on Gaussian.

This has been corrected.

- Line 120: Unless there has been a transform (e.g. ln(y)). Square bracket is facing the wrong way.
   We do not understand. Observations can be positive or equal to zero.
- Line 123: Space between units.

This has been corrected. Thank you.

- Line 156: 'Large multiple'

This has been corrected.

- Line 178: 'as this paper'

This has been corrected.

- Line 278: What are the upper/lower bounds of the uniform distribution?
   This has been added. Thank you a lot.
- Line 280: What are the shape parameters of the log-gamma distribution?
   This has been added.

- Line 348: 'Harmful' is an incorrect choice of work here. You can just delete it; This has been removed.
- Line 348-350: This sentence isn't correct. Observations don't have a high likelihood. Please rephrase.
   This has been corrected.
- Line 351: Change 'totally legitimate' to 'valid' This has been corrected.
- Line 353-355: This sentence does not make sense. I'm unsure of its meaning, please revise.
   This has been corrected. Thank you.
- Line 368: 4c and 5a

This has been corrected (4b and 5a).

- Equation A1 and A2: Second term is incorrect, not divide by 2 but multiplied by 2.

This comment has been removed by the reviewer.

Thank you very much for all these technical comments.

**References**

Liu, Y., Haussaire, J.-M., Bocquet, M., Roustan, Y., Saunier, O., and Mathieu, A.: Uncertainty quantification of pollutant source retrieval: comparison of Bayesian methods with application to the Chernobyl and Fukushima Daiichi accidental releases of radionuclides, Quarterly Journal of the Royal Meteorological Society, 143, 2886–2901, https://doi.org/10.1002/qj.3138, 2017.

---

## Author Comment (AC3)

**Discussion:** Quantification of uncertainties in the assessment of atmospheric release source with application to the autumn 2017 106Ru event**

Joffrey Dumont Le Brazidec1,2, Marc Bocquet2, Olivier Saunier1, and Yelva Roustan2 1IRSN, PSE-SANTE, SESUC, BMCA, Fontenay-aux-Roses, France 2CEREA, Joint laboratory École des Ponts ParisTech and EDF R&D, Université Paris-Est, Marne-la-Vallée, France **Correspondence:** Joffrey Dumont Le Brazidec (joffrey.dumont@enpc.fr)

**Report 3**

We thank Dr. Tichý for all his valuable comments, recommendations, and interest in the paper.

The manuscript presents interesting study based on estimation of atmospheric release from ambient concentration measurement coupled with atmospheric model. Few prior models of a release are presented, discussed and evaluated on Ruthenium 106 case from 2017. Here, there is consensus on release location and approximate release time-profile which makes this case very interesting and a playground for model testing. The manuscript is nicely written and clear to understand. What I lack is clarification and verification of some statements. I also recommend to extend conclusion (or discussion) by some suggestions and recommendations for future cases, see specific comments bellow. In sum, I would recommend the paper for publication after these clarifications.

Specific comments:

 p. 5, line 115: Although I understand the importance of lower values in measurements, there might be a good reason for high significance of higher values since they may bring more confident infromation wiht lower uncertainty, especially in case with spatialy and temporaly long transport.

We agree that the fourth criterion results from a "multiplicative" consideration of the error distribution, which is not necessarily true. However, we believe that this simplification is relevant in the case where a limited number of hyperparameters are available to model  $\mathbf{R}$ . We have specified in the text that this criterion relies on this underlying simplification.

- Figure 2: I am curious whether similar results are obtained for latitude. Considering the dominant direction of the atmospheric transport is probably in longitude axis, it is maybe different in latitude axis. Please, comment.

The difference in std for the latitude is indeed lower. We have chosen a special case where the difference is very important. Please see figure 1 for the difference between the latitude distribution with observation sorting algorithm (for

two different threshold values) and without observation sorting algorithm. In most scenarios, and for most variables, the difference between the stds of reconstructed distributions with and without the application of the sorting algorithm is about 10 to 30 %.

Figure 1. Density of the variables describing the 106Ru source for a log-Laplace likelihood with threshold  $y_t = 0.1 \text{ mBq.m}^{-3}$  of the Ruthenium source sampled with the enhanced ensemble of observation operators using the parallel tempering method with or without the help of the observation sorting algorithm. When applied, the threshold of the observation sorting algorithm is  $\epsilon_d = 0.1 \text{ mBq.m}^{-3}$  or  $0.01 \text{ mBq.m}^{-3}$ .

- p. 14, line 375: Regarding temporal profiles of the estimated release, what is exactly the time-resultion of the posterior, is it one day? Is it possible to plot the release profiles somehow, e.g. using medians or similar? Did you estimate some significant activity also in other days except 25th and 26th September?

The release is estimated as a vector of daily release rates. Each release rate on each day between the 22nd and the 28th is estimated as a 1D variable. We plot the evolution of the release rate (using medians and variances) on figure 2. No significant activity is estimated in other days except for the 25th or 26th as described in our first paper (Dumont Le Brazidec et al., 2020). There is however an exception in the case with the enhanced ensemble of observation operators, with a log-normal likelihood of threshold  $3mBq.m^{-3}$  (the coordinates of the source distribution is displaced and so is the release.)

---

## Referee Report (RR1)

I thank the authors for their patience with my previous review, which was mostly incoherent due to a formatting problem. Prior to submitting that, I submitted a referee report at the initial stage of the review process. Perhaps I submitted that prematurely, and the authors may not have seen it. That report contained the bulk of my "General and Specific Comments." Both of the old reports are included in this document in case they can still be useful to the authors, but a line-by-line response is not requested, and many of the comments have already been addressed. I am very sorry to have caused the authors so much confusion.

**June 2021 Referee Report**

**General Comments**

The manuscript is improved, particularly by clarifying the language around key points throughout, and by adding content to the summary and conclusions. I think it is fit for publication as is (so long as the technical corrections are made).

**Specific Comments**

Section 1.2: Just a suggestion, but I think a hint at what strategy you will use for weighting ensemble members is appropriate here. The reader cannot tell whether you will do something ordinary or novel.

Line 117: Your argument against a Gaussian likelihood is still a little bit wanting, in my opinion. When you say that "We think that the whole measurement set should bring information" what you are really saying is that it is more important to learn from a larger count of measurements than it is to learn from the smaller count of the largest errors. Or that relative error is what matters. Those are not self-evidently true, to me. So I think you should provide some concise statement about the benefits of learning from a larger number of observations (less sensitive to outliers?), or just say that it is worth testing.

**Technical Corrections**

Line 10: "suitable" instead of suited
Line 372: Correct typo and clarify what you mean
Line 441: Indent and expand this paragraph? Or combine it with the following paragraphs.

**Previous report (December 2020)**

This paper does exactly what it says it will do: source term estimation for an emission of Ruthenium using observations, an atmospheric model, and Bayesian inversion. It excels at explaining the concepts involved, making it especially accessible to someone who has not done this exact type of problem before. However, many corrections are needed to the wording, particularly in section 2, and the organization, particularly in section 3.

Comments:

37. In the climate modeling community we would refer to an initial conditions ensemble of the same atmospheric model as a "single-model ensemble" rather than a "multi-model ensemble."

39. Missing "a"

60-63. Comment: I like this concise explanation of how a model, source, observations, and likelihood fit together.

89. Suggested "These three sources of uncertainty are explored in an application of source term estimation for the $^{106}$Ru release…"

90.

89-94. Can you rephrase this so that it flows monotonically, i.e. reference section 2 before section 3?

103. The math is correct but the wording is not quite right. I think you mean that $r$ is a positive coefficient and R (and $r$I) is a positive diagonal matrix; $r$ itself is not a "positive diagonal coefficient."

106.

111-112. Suggest something like…*choosing Gaussian likelihood penalizes the largest errors to an extent that smaller errors are negligible.*

115. Rephrase. Consecutive sentences starting with "in other words."

116.  I think you should delete the sentence starting with ``Every " as the wording is confusing. Your example (100,120) vs (10,12) has already made this point.

120. Bracket typo.

126. What do you mean by mitigated here? I think you can say "should be 1" or "should be close to 1."

131. Missing a word here, which obscures the meaning of the sentence.

144. It may be helpful for the reader if you reference the section in which the threshold is discussed.

190. If the observation sorting algorithm is the division into $r$ and $r_{nd}$, then you should not start a new paragraph for sentence 191.

268, 274. "22$^{nd}$"

305. This summary section should be clarified if possible. For uniformity, I recommend starting each bullet point with a section number, e.g.
- Section 3.3.2 is an application of the observation sorting algorithm…;
- Section 3.3.3 is an application of the different likelihood functions and spatial clustering …;
- Section 3.3.4 is an application of the perturbed dispersion parameters and enhanced ensemble…

Secondly, the section heading "Summary" section seems out of place, especially since you have a summary section later. I would suggest renaming 3.3 "Results" and renaming 3.3.1 "Overview."

345. "Probable sources"

355. "which is not justifiable."

440. Explain when and where this accident took place, and maybe ad some thoughts about how this might compare to what you just did.

436-440. I think more discussion would be helpful for the reader. Remember, many readers skim the paper until they get to the conclusions.

**Initial Report (to determine whether manuscript should proceed to review and discussion stage)**

Initial manuscript evaluation of JDLB et al., 2020.

I very much enjoyed reading this paper. The author excels at explaining the concepts involved, which is refreshing. For example, the intuition of how the likelihood functions affect posteriors, etc., is well-done. The results are presented clearly.

There are a handful of language errors that should be addressed before a preprint is posted on the web. I have documented some of them here but would strongly recommend another proofread. **Any comment that should addressed before preprint is in bold.** General comments more suited to a full review are also included, but those do not need to be addressed at this stage.

**1. Typo: "a an" -> an**
**2. Language: "fruitful in  recent years."**

61. Comment: I particularly like this explanation of how a model is used as a tool to calculate the likelihood.

Equation 2. Why the '...'?

**105. Language: When I read "Inverse" I think of a matrix inverse or ^(-1). I believe the author means "negative" in this context. If so, this is very important to fix.**

109. Comment: In the case where the observations and the predictions are all equal, is equation 4, the cost, actually equal to zero? To me it looks like the answer is no.

112. The cost of couple 1 is 100x larger than couple 2 only if couple 1 and couple 2 are assumed to have the same variance, so this isn't necessarily a property of the Gaussian distribution itself.

179-186. Comment: This paragraph is generally well-written and the concept is clear.

**273. I do not understand the meaning of "impulsions" in this context. If this is a technical term then I apologize. If this is a language error, I think the author is referring to the daily average release rate, so "releases" or "emissions" would suffice.**

298. The variances of these transition parameters is listed here, but what about the other parameter for the folded normal distribution? Is there not a "location parameter?" I cannot tell if that is specified somewhere. Perhaps I am misunderstanding.

Fig 4 Caption: I think this caption could be rewritten to be more clear, even if it takes more text.

Fig8. Comment: Remove the fill here so we can see more clearly?

Fig. B1. Is it possible to overlay where your ensembles fit on this plot? Otherwise it is a little eyebrow-raising to say that the criteria "totally justify" using this method, as you do on line 470. Also, what are the implications of appendix B for using three spatial clusters? This seems worth a comment.

474. Could you clarify what is a hit and what is a false occurrence? This may be obvious to some readers but not to me. Is a "hit" correctly identifying the bin of the observation?

---

## Author Response (AR2)

**Report 1**

« I am happy with the revision and think that this paper will make a valuable contribution. I congratulate the authors on their work.

My only comment is that, in the revised manuscript, the title is no longer grammatically correct. I suggest:
Quantification of uncertainties in the assessment of *an* atmospheric
release source *applied* to the autumn 2017 106Ru event »

The title has been modified. We thank again for his/her kind words, constructive comments, and technical corrections the reviewer.

**Report 2**

« Very nice revision of the manuscript, see few technical comments:

p. 20, l. 442 and 445: I suggest to use "the first step" and "the second step".

p. 20, l. 449: Finally, in order to incorporate into the sampling process the uncertainties related to the meteorological fields and the transport model,... --> Finally, in order to incorporate the uncertainties related to the meteorological fields and the transport model into the sampling process,...

p. 22, l. 461: missing i in "meteorologcal"

p. 29, l. 647: Please, cite the published version, see https://doi.org/10.5194/amt-14-803-2021 »

Technical corrections have been applied. We thank a lot again for his/her valuable comments, technical corrections and recommendations the reviewer.

**Report 3**

We thank again for his comments Dr. De Meutter.

**Report 4**

« The manuscript is improved, particularly by clarifying the language around key points throughout, and by adding content to the summary and conclusions. I think it is fit for publication as is (so long as the technical corrections are made).

Specific Comments
Section 1.2: Just a suggestion, but I think a hint at what strategy you will use for weighting ensemble members is appropriate here. The reader cannot tell whether you will do something ordinary or novel. »

We have added :
"We proposed in this paper a new technique to estimate the weights associated to the ensemble members."

« Line 117: Your argument against a Gaussian likelihood is still a little bit wanting, in my opinion. When you say that "We think that the whole measurement set should bring information" what

you are really saying is that it is more important to learn from a larger count of measurements than it is to learn from the smaller count of the largest errors. Or that relative error is what matters. Those are not self-evidently true, to me. So I think you should provide some concise statement about the benefits of learning from a larger number of observations (less sensitive to outliers?), or just say that it is worth testing. »

We have added
"The whole set of measurements should provide information: if the inversion is dominated by the few measurements with the largest errors (which may possibly be outliers), valuable information provided by the other measurements may be missed."

« Technical Corrections
Line 10: "suitable" instead of suited
Line 372: Correct typo and clarify what you mean
Line 441: Indent and expand this paragraph? Or combine it with the following paragraphs. »

Technical corrections have been made. Thanks a lot. Previous comments have been checked and taken in account as well.
For line 372, we have added
"Therefore, the variances are only representative of observations with high uncertainties in their subset."

We thank again the reviewer for his/her comments/revisions that improved the quality of this manuscript.